# Validation of Gait Measurements on Short-Distance Walkways Using Azure Kinect DK in Patients Receiving Chronic Hemodialysis

**DOI:** 10.3390/jpm13071181

**Published:** 2023-07-24

**Authors:** Zhi-Ren Tsai, Chin-Chi Kuo, Cheng-Jui Wang, Jeffrey J. P. Tsai, Hsin-Hsu Chou

**Affiliations:** 1Department of Computer Science & Information Engineering, Asia University, Taichung 41354, Taiwan; 2Department of Medical Research, China Medical University Hospital, Taichung 404327, Taiwan; 3Center for Precision Medicine Research, Asia University, Taichung 41354, Taiwan; jjptsai@gmail.com; 4Department of Bioinformatics and Medical Engineering, Asia University, Taichung 41354, Taiwan; 5Department of Pediatrics, Ditmanson Medical Foundation Chia-Yi Christian Hospital, Chiayi 600, Taiwan

**Keywords:** Kinect v3 gait measurement and analysis system (K3S), forward angle, chronic dialysis

## Abstract

Muscle dysfunction, skeletal muscle fibrosis, and disability are associated with weakness in patients with end-stage renal disease. The main purpose of this study was to validate the effectiveness of a proposed system for gait monitoring on short-distance 1.5 m walkways in a dialysis center. Gaits with reduced speed and stride length, long sit-to-stand time (SST), two forward angles, and two unbalanced gait regions are defined in the proposed Kinect v3 gait measurement and analysis system (K3S) and have been considered clinical features in end-stage renal disease (ESRD) associated with poor dialysis outcomes. The stride and pace calibrations of the Kinect v3 system are based on the Zeno Walkway. Its single rating intraclass correlation (ICC) for the stride is 0.990, and its single rating ICC for the pace is 0.920. The SST calibration of Kinect v3 is based on a pressure insole; its single rating ICC for the SST is 0.871. A total of 75 patients on chronic dialysis underwent gait measurement and analysis during walking and weighing actions. After dialysis, patients demonstrated a smaller stride (*p* < 0.001) and longer SST (*p* < 0.001). The results demonstrate that patients’ physical fitness was greatly reduced after dialysis. This study ensures patients’ adequate physical gait strength to cope with the dialysis-associated physical exhaustion risk by tracing gait outliers. As decreased stride and pace are associated with an increased risk of falls, further studies are warranted to evaluate the clinical benefits of monitoring gait with the proposed reliable and valid system in order to reduce fall risk in hemodialysis patients.

## 1. Introduction

Gait abnormalities, including slow gait speed and decreased stride length, are prevalent in chronic kidney disease (CKD) patients, especially in end-stage renal disease (ESRD) patients receiving chronic hemodialysis [1,2], excluding patients with acute kidney injury [3]. Gait impairment predisposes patients to falls and related consequences and is associated with increased hospital admission, cognitive dysfunction, cardiovascular events, and all-cause mortality [4,5,6].

Skeletal muscle fibrosis impairs muscle function and is considered a major cause of muscle weakness [7]. Moreover, the isometric leg extension strength and measures of skeletal muscle fibrosis in CKD patients were examined in a previous study [8] which supported the conclusion that skeletal muscle fibrosis is associated with weakness in patients with CKD. Moreover, ESRD accelerates brain neurological function degeneration, thus further affecting gait function. Hence, gait analysis of ESRD patients is a crucial component of alerting medical personnel to identify and treat possible underlying disorders which arise from neurological diseases, including cerebral palsy and multiple sclerosis [9,10,11]. Monitoring the walking gait in ESRD patients is necessary for the early identification of patients suffering from skeletal muscle fibrosis, which is a disease that causes muscle weakness. In addition, serial gait analysis can be used as a marker to monitor the effects of treatment. Various rehabilitation practitioners use gait analysis to check the effects of treatment [12].

Previously, controlled clinical gait analysis was commonly performed with timed walking tests utilizing a long-distance (≥5 m) pressure-sensitive Zeno Walkway [10] or a Vicon motion capture system made out of 10 T40 cameras, thus requiring a large surrounding space [13,14]. The GAITRite, Zeno ProtoKinetics, OptoGait, and Vicon systems are also expensive and require significant setup time, expert knowledge, specialized locations, and precise calibration to extract gait features reliably for movement disorder rehabilitation and are thus not suited for use in noncontrolled clinical settings. In addition, the experimental protocols of GAITRite and Zeno pressure-sensitive walkways demand a 5+ m walkway and are unable to directly measure the sit-to-stand time (SST). Walking trials using the CPU-based Kinect v1–v2 [15,16,17,18,19,20,21] together with the Zeno Walkway were demonstrated in several studies [10,22,23,24,25,26,27,28]. Additionally, the methodology for use of Kinect v1–v2 calibrated using the Vicon motion capture system was revealed in a few studies [13,14]. According to previous research [29], the obvious advantages of the GPU-based Azure Kinect (Kinect v3) compared with the CPU-based Kinect v1–v2 were as follows: (1) better angular resolution; (2) lower noise; (3) good accuracy. In addition, a gait with slowed speed, reduced stride length, and poor balance has been considered a clinical feature in the GaitBEST gait analysis software [30], which was developed on a long-distance 4 m walkway and Kinect v1–v2 with a timed method used to provide stride, pace, and fall risk. However, discontinuation or loss of gait monitoring due to the occlusion of the target’s skeleton frame by other people may be encountered in some fields with limited space, as for example, in the dialysis center. A short-distance 1.5 m walkway is more suitable for setting-up a gait measurement system. Moreover, to avoid overwhelming patients, the proposed system included a long-term automated gait measurement; a nontimed, self-service, and unassisted method; reasonable segmentation of gait data (Virtual Skeleton Frames, VSF); and an automated recognition method for patients. Hence, we proposed a new gait system, the Kinect v3 gait measurement and analysis system (K3S), which was calibrated using the Zeno Walkway to evaluate its reliability and validity. The Zeno Walkway provides ground truth data for validation purposes and is not part of the proposed system. Moreover, it meets these requirements for self-selected and unguarded gait tests and supports the six gait parameters of calculations: maximum stride (MS), average pace (AP), SST, dip walk forward angle (DWF), dip sit-to-stand forward angle (DSSF), and unbalanced point number (UPN) in the two unbalanced gait regions. This UPN is extended from single-foot balance theory [31], but it is calculated without the base of support (BOS), and it is not similar to previous research [30]. Currently, the incidence of hunchback, cervical spondylosis, lumbar spondylosis, and other diseases caused by poor sitting posture is increasing, which has a serious impact on people’s learning and life [32]. Hence, this study proposes that DWF and DSSF be used to replace the different monitoring methods revealed by the paper [32].

## 2. Gait Analysis for CMUH Patients

### 2.1. Subjects

The flowchart of patient selection for this gait experiment for the self-selected gait tests of every participant using three locations (walkways 1–3) in the Dialysis Center of CMUH is shown in Figure 1, and the baseline demographic data are described in Table 1. In total, 3281 valid data points (1461 before dialysis and 1115 after dialysis) from 75 chronic dialysis patients were collected from 12 November 2021 to 18 May 2022. The median age of participants was 68 years (IQR 59.5, 72.3 years) with male predominance (65.3%).

### 2.2. Study Procedures

A clinical trial using three locations of the Dialysis Center of China Medical University Hospital (CMUH) was implemented to conduct unguarded gait monitoring of three independent 1.5 m walkways. This clinical trial was approved by the Institutional Review Board (CMUH108-REC2-022) of CMUH Research Ethics Center. In addition, all methods performed in this study were in accordance with current regulations and guidelines.

There is a four-step workflow shown in Figure 2 for the proposed Kinect v3 gait measurement and analysis system (K3S). These four steps are as follows:

Step 1: The K3S detects the RFID tag ID when a user uses an RFID tag to approximate an RFID reader of an Arduino chip connected to this system by a wired connection, and the user is guided by an RGB LED. Then, go to Step 2;

Step 2: The K3S checks whether this ID’s user is allowed to measure this user’s gait. If this ID is allowed, then the K3S starts to measure the hand grip strength of the hand dynamometer (optional) or goes to Step 3;

Step 3: Turn on the Kinect v3 automatically for patients with RFID tags to trigger the skeleton frame measurement function and go to Step 4;

Step 4: The Kinect v3 identifies the user that sits on a specific weighing chair and then tracks this user’s VSF trajectory on the walkway and chair. When the user leaves this walkway, the K3S starts analyzing the VSF, which provides 32 virtual skeleton joints labeled No. 1–No. 32, as shown in Figure 2, to generate the six gait parameters (maximum stride, average pace, SST, DWF, DSSF, UPN).

### 2.3. Calibration of Walkways

The Kinect v3 coordinate needs to be calibrated by the proposed calibration shown in Figure 3 at the Asia University (AU) laboratory. Hence, Kinect v3′s coordinates of walkways 1–3 for the same participant to a unified coordinate system are calibrated using the following coordinate transformation, and its calibration result for K3S is shown in Figure 3: θ = atan(·) is the function for calculating the inverse tangent in radians as θ. The output = sign(input) is the function of calculating an array output the same size as input, where each element of output is designated as 1 if the corresponding element of input is greater than 0; 0 if the corresponding element of input equals 0; −1 if the corresponding element of input is less than 0. The calibration steps are as follows:Kinect v3 captures the skeleton frames of the participant walking from the starting point to the ending point on a short-distance 1.5 m walkway and goes to Step 2;The 2D grid coordinates are based on the coordinates contained in vectors x and y of the foot joints, which are in these skeleton frames. X is a matrix where each row is a copy of x, and Y is a matrix where each column is a copy of y. The grid represented by the coordinates X and Y has length (y) rows and length (x) columns. The grid size is 1 cm. Go to Step 3;Calculate z^ to fit a surface of the form z = f(x, y) to the scattered data in the vectors (x, y, z) of foot joints in these skeleton frames while the function f interpolates the surface at the query points specified by (X, Y) and returns the interpolated values, z^. The surface always passes through the data points defined by x and y. Go to Step 4;Compute the angle θ_y_ = atan((max(z) − min(z))/(max(x) − min(x))) · 80/π·sign(z(arg[min(x)]) − z(arg[max(x)]));y_11_ = cos(θ_y_ · π/180); y_13_ = sin(θ_y_ · π/180); y_31_ = −sin(θ_y_ · π/180); y_33_ = cos(θ_y_ · π/180);
(x, y, z) ← (X,Y,z^)·y110y13010y310y33;
$$\mathrm{No}.22(x,\;y,\;z)\;\leftarrow \;\mathrm{No}.22(x,\;y,\;z)\cdot \left[\begin{array}{ccc} y_{11} & 0 & y_{13}\\ 0 & 1 & 0\\ y_{31} & 0 & y_{33} \end{array}\right];$$
$$\mathrm{No}.26(x,\;y,\;z)\;\leftarrow \;\mathrm{No}.26(x,\;y,\;z)\cdot \left[\begin{array}{ccc} y_{11} & 0 & y_{13}\\ 0 & 1 & 0\\ y_{31} & 0 & y_{33} \end{array}\right].$$

Go to Step 5;
5.(b, a) ← the maximum difference of (No.22 (x, y)+ No.26 (x, y))/2. Go to Step 6;6.θ_z_ = atan(a/b); z_11_ = cos(θ_z_); z_12_ = −sin(θ_z_); z_21_ = sin(θ_z_); z_22_ = cos(θ_z_);
$$\mathrm{The}\;\mathrm{calibration}\;(x,\;y,\;z)\;\leftarrow \;(x,\;y,\;z)\cdot \left[\begin{array}{ccc} z_{11} & z_{12} & 0\\ z_{21} & z_{22} & 0\\ 0 & 0 & 1 \end{array}\right].$$

**Figure 3 jpm-13-01181-f003:**
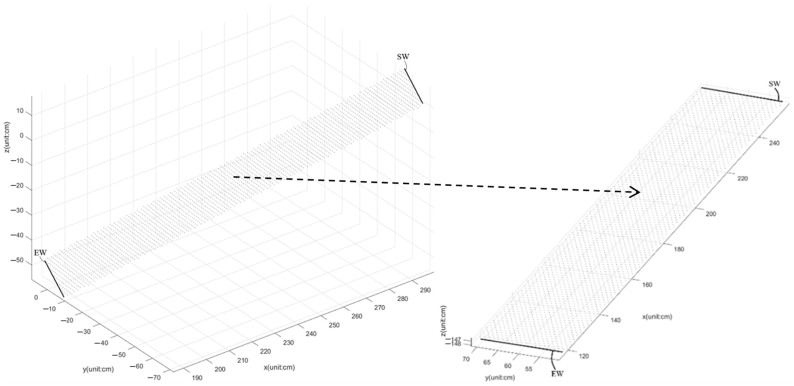
Before coordinate transformation, the interpolation plane of the virtual foot coordinates was generated using No. 22 (x, y, z) and No. 26 (x, y, z) of Figure 2 in the region of the ground plane from SW to EW. The calibrated ground plane was obtained after coordinate transformation.

### 2.4. Detection and Correction of Unreasonable Gait Movement

First, the K3S experiments encountered some unreasonable movements, such as suddenly stopping and bending over to pick up some objects, which is shown in Figure 4a; walking back and forth, which is shown in Figure 4b; sitting in a wheelchair to move on the walkway, which is shown in Figure 4c; or a generalized stride, which is extended from Figure 4e and shown in Figure 4f.

This study identifies and detects these unreasonable movements on the walkway using the proposed mechanism with some functions as the following one-way trajectory detection. The result of selecting the two reasonable one-way trajectories for the walking case of Figure 4b is shown in Figure 4d.

The purpose of this detection is to select the two reasonable one-way trajectories, the left ankle one-way trajectory a6, and the right ankle one-way trajectory a7, from various walking and interfering trajectories. The joint filter design involved local k-point averages using the moving mean method (Mm). Each average is calculated over a sliding window of length k that spans the position array’s joint coordinates of the time series adjacent elements. When k is odd, the window is centered on the element at the current position. When k is even, the window is centered on the current and previous elements. The window size is automatically truncated at the endpoints when there are not enough elements to fill the window. When the window is truncated, the average calculation only takes the elements that fill the window, so the array length is the same before and after filtering. This study uses k = 5. The diff function calculates the difference between adjacent elements of an array. The one-way trajectory detection algorithm is as follows:If the height of the skeleton is <61 cm, then stop this algorithm or else, go to Step 2;If the length of skeleton frames >31, then a unit of sampling time = 0.04 s and a suitable moving window (units of sampling time × the range of ankle(x) cm) are used to find the longest segment (LS) of the skeleton frames (SF) and calculate the average pace (AP) = the length of pelvis(x) trajectory of LS/the duration of LS, then go to Step 3. Otherwise, stop this algorithm;Calculate Mm21 ← Mm(No.21(x)), Mm25 ← Mm(No.25(x)), a1 ← (Mm21 + Mm25)/2, go to Step 4;(b1 ← arg(a3)) ← (a3 ← max(a1)), go to Step 5;(b2 ← arg(a4)) ← (a4 ← min(a1)), go to Step 6;If (b1 > b2), then (c1 ← 0), (s1 ← b1), (a5 ← a1(b1)), go to Steps 7, 8, and 9;For i ∈ [b1,b2] do:If (a1(i) ≤ a5), then (a5 ← a1(i)), (s1 ← i), (c1 ← 0), else (c1 ← c1 + 1);If (c1 > 10), then break the for-loop, (a6 ← No.21(x([s1,b1]))), (a7 ← No.25(x([s1,b1])));Else (c1 ← 0), (s1 ← b1), (a5 ← a1(b1)), go to Steps 11, 12, and 13;For i ∈ [b1,b2] do:If (a1(i) ≤ a5), then (a5 ← a1(i)), (s1 ← i), (c1 ← 0), else (c1 ← c1 + 1);If (c1 > 10), then break the for-loop, (a6 ← No.21(x([b1,s1]))), (a7 ← No.25(x([b1,s1])));a2 ← a6-a7; a8 ← max{|diff(a6)|,|diff(a7)|};If a8 > 50, then stop this algorithm or else, go to Step 16;Calculate the maximum stride (MS), then stop this algorithm.

### 2.5. Calibration of the Forward Angle

The forward angle θ of K3S analysis needs to be calibrated by an online protractor (https://www.ginifab.com/feeds/angle_measurement/, accessed on 15 July 2022) to measure the real θ. Hence, the proposed K3S forward angle θ = atan(b/a)·180/π, where a = norm(pelvis(x, y, z) − neck(x, y, z)); b = norm(pelvis2(x, y, z) − neck(x, y, z)); pelvis2(x, y, z) = (pelvis(x), pelvis(y), neck(z)); norm(·) is a function for calculating the Euclidean norm (also known as the 2-norm) of the vector.

### 2.6. Detection of Personal DWF and DSSF Outliers

The median ± standard deviation (SD) of θ (°) were used as the upper and lower boundaries, respectively, and they were used to detect the personal DWF and DSSF outliers based on personal statistics. For example, the boundary design of the DWF for patient 1 is shown in Figure 5a. The DWF was an outlier if the DWF > upper boundary. The boundary design of the DSSF for patient 1 is shown in Figure 5b. The DSSF was the outlier if DSSF < lower boundary.

### 2.7. Data of Personal Unbalanced Gait

The proposed personal unbalanced gait for UPN on every gait test in the two unbalanced gait regions was extended from pendulum model theory [31] and was based on the personal statistic boundaries (median ± SD), i.e., as Figure 6 for patient 1 to quantify his fall risk degree. The UPN is designed as follows:

**Step 1:** Calculate the personal CM(x) along the *x*-axis, where
CM(x, y, z) ≡ g1C1 + g2C2 + ⋯ + g15C15,
gi ≡ Ai·Di + Bi·(1 − Di),
i = 1~15. The Ai, Bi, Di, gi, and Ci parameters are listed in Table 2;

**Step 2:** Calculate the median ± SD of the phase portrait of personal statistics for CM(x)−No. 22(x) and CM(x)−No. 26(x) to be the lower boundary, upper boundary, left boundary, and right boundary;

**Step 3:** Count the outliers in the green and yellow boxes for the phase portrait of CM(x)−No. 22(x) and CM(x)−No. 26(x) to be UPN on every gait test in the two unbalanced gait regions for the personal boundaries.

Hence, this analysis with a center of mass (CM) of a VSF is proposed to calculate the personal UPN in the personal two unbalanced gait regions based on the phase portrait. The two regions are calculated using median ± SD boundaries of the personal phase portrait based on the CM. These unbalanced points’ definition is from the two unbalanced gait regions and is extended from pendulum model theory [31] to quantify the fall risk.

### 2.8. Calibration of the Sit-and-Stand Time (SST) Algorithm of K3S

The sit-and-stand time (SST) of K3S analysis needs to be calibrated using a pressure insole with two sensors A and B to measure the higher frame-rate SST (Figure 7).

Three participants are required to test the two sensors (A and B) of a pressure insole to calibrate the following algorithm of K3S:

Kinect v3 is fixed to the wall that is set as x = 0 cm, and a chair is placed in the range 0 cm < x < 90 cm.
If the height of the skeleton is <61 cm, then stop this algorithm or else, go to Step 2;If the length of skeleton frames >31, then go to Step 3. Otherwise, stop this algorithm;If 0 < No.1(x) < 90 cm, then define that as the range of calculating SST. Go to Step 4.A unit of sampling time = 0.04 s and a suitable moving window (units of sampling time × the range of pelvis(z) cm) are used to find the longest segment of the skeleton frames (SF) to calculate the SST. Go to Step 5;If this segment of SF number <33, then stop this algorithm, or else, go to Step 6;Use the moving mean method (Mm) to filter pelvis(z) trajectory No. 1(z) of SF. Go to Step 7: Find local maxima points and local minima points of No. 1(z). Go to Step 8;If there are more than two neighboring local maxima points, then select the leftmost local maxima point to be reserved. Go to Step 9;If there are more than two neighboring local minima points, then reserve the rightmost local minima point t1, which has to satisfy the condition No. 1(z) of this point ≤ min(No. 1(z)) + 10 cm, and find its neighboring local maxima point t2. Go to Step 10;If there is only one local minima point and it is the initial point of No. 1(z), then find the reasonable corner point t1 from the range between the initial point and the neighbor local maxima point t2. Go to Step 11;Select the most reasonable data range No. 1(z*) that has the maximum difference of No. 1(x), which is at least 1.5 cm long. Go to Step 12;If the difference in No. 1(z*) > 6 cm, then SST≡t2 − t1; otherwise, SST does not exist.

Next, pressure sensor A was placed under a chair’s foot while a user was sitting on the chair, and his foot was prepared to stand on pressure sensor B. After these two pressures were converted into a voltage signal with high frequency noise, they were only used through the local 10-point-average filter using the moving mean method (Mm). Next, the SST calibration experiment results for three male subjects (1–3) under 50 years old in the AU laboratory are shown in Figure 8 where male subject 2 has a height of 1.7 m and male subject 3 has a height of 1.76 m.

## 3. Results

### 3.1. K3S Calibration Results

We used the same calibration method to obtain walkways 1–3 for the Kinect v3 set at points A, B, C, D, and its blind spots, which are shown in Figure 9. Moreover, it was used to conduct short-distance 1.5 m walkways and nontimed gait monitoring experiments. Our gait experimental system consisted of the Kinect v3 devices, which were set for three locations as follows, and their VSF was running at 25~30 Hz:

Walkway 1: The A, B, and C corner points, as shown in Figure 9a, of the Kinect v3 were positioned as follows: A: 180 cm, B: 172 cm, and C: 171.5 cm above the ground; 52 cm was the length from D to chair center projection x value along the *x*-axis, and 28 cm was the length from chair center projection y value to D along the *y*-axis. The location is shown in Figure 9a.
$$\left[\begin{array}{ccc} y_{11} & 0 & y_{13}\\ 0 & 1 & 0\\ y_{31} & 0 & y_{33} \end{array}\right]=\left[\begin{array}{ccc} 0.738 & 0 & 0.6748\\ 0 & 1 & 0\\ -0.6748 & 0 & 0.738 \end{array}\right];$$
$$\left[\begin{array}{ccc} z_{11} & z_{12} & 0\\ z_{21} & z_{22} & 0\\ 0 & 0 & 1 \end{array}\right]=\left[\begin{array}{ccc} 0.992 & -0.126 & 0\\ 0.126 & 0.992 & 0\\ 0 & 0 & 1 \end{array}\right].$$

Walkway 2: The B, C, and D corner points, as shown in Figure 9b, of Kinect v3 were positioned as follows: B: 207 cm, C: 196 cm, and D: 197 cm above the ground; 28 cm was the length from C to chair center projection x value along the *x*-axis, and 92 cm was the length from chair center projection y value to C along the *y*-axis. The location is shown in Figure 9b.
$$\left[\begin{array}{ccc} y_{11} & 0 & y_{13}\\ 0 & 1 & 0\\ y_{31} & 0 & y_{33} \end{array}\right]=\left[\begin{array}{ccc} 0.7112 & 0 & 0.703\\ 0 & 1 & 0\\ -0.703 & 0 & 0.7112 \end{array}\right];$$
$$\left[\begin{array}{ccc} z_{11} & z_{12} & 0\\ z_{21} & z_{22} & 0\\ 0 & 0 & 1 \end{array}\right]=\left[\begin{array}{ccc} 0.8296 & 0.5584 & 0\\ -0.5584 & 0.8296 & 0\\ 0 & 0 & 1 \end{array}\right].$$

Walkway 3: The B, C, and D corner points, as shown in Figure 9c, of Kinect v3 were positioned as follows: B: 201 cm, C: 193 cm, and D: 193 cm above the ground; 49 cm was the length from D to chair center projection x value along the *x*-axis, and 20 cm was the length from chair center projection y value to D along the *y*-axis. The location is shown in Figure 9c.
$$\left[\begin{array}{ccc} y_{11} & 0 & y_{13}\\ 0 & 1 & 0\\ y_{31} & 0 & y_{33} \end{array}\right]=\left[\begin{array}{ccc} 0.6285 & 0 & 0.7778\\ 0 & 1 & 0\\ -0.7778 & 0 & 0.6285 \end{array}\right];$$
$$\left[\begin{array}{ccc} z_{11} & z_{12} & 0\\ z_{21} & z_{22} & 0\\ 0 & 0 & 1 \end{array}\right]=\left[\begin{array}{ccc} 0.9995 & 0.0322 & 0\\ -0.0322 & 0.9995 & 0\\ 0 & 0 & 1 \end{array}\right].$$

### 3.2. Calibration Results of the Stride and Pace

The stride and pace of K3S analysis need to be calibrated by a gold-standard device, the Zeno Walkway, to measure the stride and pace with higher accuracy and verify the reliability and validity of the K3S. Moreover, the calibration setting is shown in Figure 10 with the four strides and their paces measured by the Zeno Walkway and K3S.

The calibration result for the stride and pace of the K3S analysis is shown in Figure 11. Its single rating intraclass correlation (ICC) for the stride with a one-way model is 0.99 (Type: absolute agreement; subjects = 40; F Test, H0: r0 = 0; H1: r0 > 0; *p* = 1.21 × 10^−35^); 95% confidence interval (CI) 0.981–0.995). The single rating ICC for the pace with a one-way model is 0.92 (Type: absolute agreement; subjects = 40; F Test, H0: r0 = 0; H1: r0 > 0; *p* = 4 × 10^−18^); 95% CI 0.855–0.957).

### 3.3. Stability Results of the Main Virtual Joints

The five adhesive tape lines were used as the markers shown in Figure 12a in the AU laboratory together with a tape measure to mark and position the main virtual joints (foot, ankle, pelvis, neck) shown in Figure 12b while walking ten times on a short-distance walkway to measure the uncertainty errors. The uncertainty errors (e_u_) are defined as e_u_ ≡ v_m_ − v_f_, where the *x*-axis coordinates v_m_ of these main virtual joints of a VSF were measured for a participant standing on the three positions (adhesive tape lines placed at 40 cm, 80 cm, and 120 cm positions, respectively) shown in Figure 12c with three bending poses from 5 to 60 degrees. He posed for the unchanged standing and bending poses as the desired *x*-axis values v_f_ measured by this tape measure. In this study, the e_u_(foot) range of the virtual foot joints was −1.58 cm ≤ e_u_ ≤ 0.66 cm; the e_u_(ankle) range of the virtual ankle joints was −0.54 cm ≤ e_u_ ≤ 1.06 cm; the e_u_(pelvis) range of the virtual pelvis joint was −0.56 cm ≤ e_u_ ≤ 1.04 cm; and the e_u_(neck) range of the virtual neck joint was −0.24 cm ≤ e_u_ ≤ 0.46 cm. Hence, the uncertainty error range of the virtual ankle joints was smaller than that of the virtual foot joints. Due to the limitation of walking on a short-distance 1.5 m walkway and setting Kinect v3, a skew angle pose has to be fixed on the wall to cover the optimal measuring region. Occluding objects and chairs on walkways 1–3 shown in Figure 9 are for the blind spots solved by choosing the stable pelvis joint to cover the noises of blind spots or occluding objects.

### 3.4. Analysis of Reliability and Validity of Forward Angle

In Figure 13, the K3S forward angle θ error range is −2.3212~0.8432°.

### 3.5. Analysis of Reliability and Validity of SST

The SST calibration experiment results for subjects 1–3 are shown in Figure 14. The sit-to-stand time (SST) calibration of Kinect v3 is based on a pressure insole and its single rating ICC for the SST with a one-way model is 0.871 (Type: absolute agreement; subjects = 17; F Test, H0: r0 = 0; H1: r0 > 0; *p* = 6.84 × 10^−7^); 95% CI 0.686–0.951).

### 3.6. Analysis of Gait Statistics

The statistical analysis of the data of 75 patients was obtained from long-term monitoring of the six gait parameters on short-distance walkways using K3S. The detection of DWF outliers, DSSF outliers, and UPN was based on the personal median ± SD boundaries. Table 3 shows the median ± SD, median (P5, P95) and the median (P1, P99) outliers of the maximum stride (MS), average pace (AP), and SST data of all participants. Table 4 shows the number of outliers of any of the three gait parameters. Furthermore, this study proposes median ± SD boundaries, median (P5, P95), and median (P1, P99) as the following methods to define the warning outliers of gait parameters to cope with dialysis physical exhaustion risk. First, the warning stride outliers were defined below the median–SD boundary of maximum stride data; their distributed numbers are shown in Figure 15a for all participants. Second, the maximum stride lengths for all participants were related to their height histogram, which are shown in Figure 15b. Hence, maximum stride length was normalized to the subject’s height using the following formula:Normalized maximum stride length = Maximum stride length (cm)/height (cm),
where its warning outliers were defined below the median–SD boundary of the normalized maximum stride length data; their distributed numbers are shown in Figure 15c for all participants. Third, the warning pace outliers were defined below the median–SD boundary of average pace data; their distributed numbers are shown in Figure 15d for all participants. Fourth, the warning SST outliers were defined beyond the median + SD boundary of the SST data; their distributed numbers are shown in Figure 15e for all participants. Fifth, the DWF outliers were defined beyond the median + SD boundary of personal DWF data. To obtain the warning DWF outliers beyond a boundary, Figure 15f shows the distributed numbers of the DWF outliers of all participants to set this median + SD boundary. Sixth, the warning DSSF outliers were defined below the median–SD boundary of the personal DSSF data;
$$\frac{The\;warning\;DSSF\;outliers\;number}{Total\;DSSF\;data\;amount}=28.0071\%$$
of participants needed an intervention to improve their DSSF. Seventh, the UPN was counted in the two unbalanced gait regions with the median ± SD boundaries of personal phase portrait data based on the personal dynamic CM data. The distributed numbers of the UPN of all participants are shown in Figure 15g to define the warning UPN outliers, which are defined beyond the median + SD boundary of these distributed numbers to alert its warning.

The adequacy of the physical gait strength of chronic dialysis patients could be detected based on the results of Figure 15, Table 3 and Table 4, that is, to cope with dialysis physical exhaustion risk by tracing the gait outliers. Furthermore, Table 5 revealed that patients after dialysis have a smaller stride (pre-test: 56.2 ± 15.7 cm, post-test: 85.0 ± 16.7 cm, *p* < 0.001), comparable pace (pre-test: 56.2 cm/s, post-test: 55.7 ± 16.9 cm/s, *p* = 0.409), and longer SST (pre-test: median 2.8, IQR 1.9, 4.5; post-test: median 3.6, IQR 2.3, 5.7; *p* < 0.001). The results were consistent with our clinical experience and demonstrate that patients’ physical fitness might be greatly reduced by dialysis.

## 4. Discussion

In this study, we developed a short-distance 1.5 m walkway for gait monitoring based on a Kinect v3 gait measurement and analysis system. Calibration algorithms for stride, pace, and SST were proposed based on a Zeno Walkway which yielded satisfactory correlations. Patients on chronic dialysis who received the short-distance gait monitoring system revealed significant differences on gait parameters before and after dialysis, including the maximum stride and SST. This system shows a great advantage in patients gait monitoring especially in the medical fields with limited space, such as a hemodialysis center or a geriatric ward. It is difficult for most hemodialysis centers to implement a traditional 4 m Zeno Walkway. Our short-distance 1.5 m walkway is more practical in clinical settings. Moreover, chronic dialysis patients are usually unable to walk as requested when asked to walk at the personal fastest speed, which is the standard protocol in the clinical setting of dialysis centers; hence, for the self-selected gait case, this study uses more accurate ankle positions and a reasonable one-way trajectory to calculate the most acceptable gait speed and stride length. Additionally, it matches the height to judge whether it is a wheelchair case to automatically discard the invalid gait speed and stride length for that case.

Taking the 60–64 age group as an example of comprehensive geriatric assessment (CGA), the gait speed of 0.55 ± 0.14 m/s that is shown in Figure 15d and the SST time of 1.93 ± 1.73 s that is shown in Figure 15e of K3S compared to CGA and GST-4 (gait speed test-4 m), mean that the 4 m test should be completed within 5 s and that the converting speed is 0.8 m/s. Moreover, the standard SST value of each time required is approximately 30 s/28 times = 1.07 s for men and approximately 30 s/24 times = 1.25 s for women using CGA. Since most dialysis patients cannot undergo 5XSST or CGA in the dialysis center, this study found the most reasonable method to detect SST. Furthermore, a stride length of 0.7 ± 0.15 m from the K3S, as shown in Figure 15a, is comparable to the stride length of 1.05 ± 0.22 m from 9 ESRD patients aged 60–74 [2]. In this study, our subjects had a lower height, needed to walk on a shorter distance walkway, and had the obstacle of a chair, so their gait speed and stride length were lower than normal. The SST was longer because of relatively weak physical strength or the distracting effect of weighing.

Although our results are promising, we did not directly compare the enrolled chronic dialysis patients with short-distance walkways using the K3S with a Zeno Walkway, which makes the correlation between them in chronic dialysis patients unknown. It is possible that the speed detected on the short-distance walkway might be slower than the longer-distance walkway due to a self-selected pace with a short distance for speeding up during walking. Therefore, we proposed the concept of detection of abnormal gait based on outliers in gait parameters after normalization, according to the subject’s height. Future studies should focus on the validation of the utility and effectiveness of gait parameter outliers on a short-distance walkway.

The reason for limiting the study subjects to chronic dialysis patients is that this system is designed to cater to the needs of nephrologists. However, its versatility is excellent, allowing for potential extension to the application areas of long-term care centers or rehabilitation departments. All that is required is recalibration and redefinition of the warning threshold for use.

## 5. Conclusions

This study proposed an alternative K3S suited for three short-distance 1.5 m clinical gait walkways. A generalized stride length is computed using the more stable virtual ankle joints rather than the virtual foot joints, and the pace is calculated using the more stable pelvis joint. The main contributions of this study are as follows: (1) The Kinect v3 virtual skeleton frames of gaits are calibrated using a Zeno Walkway and a pressure insole, and the estimation errors of some virtual joints are calculated for three experimenters using a tape measure and an image-based protractor to increase awareness of system accuracy and robustness; (2) The nontimed measurement and the clinical gait analysis of the whole body for self-selected gaits are based on a filter design and can be utilized for long-term monitoring application for chronic dialysis patients; (3) A method is proposed to simulate the instability range of a single footstep based on the center of mass calculated using virtual skeletal joints; (4) This study proposes trunk forward angles of walking and sit-to-stand posture to accurately detect abnormal changes in human posture to provide an early notification of an unhealthy posture to patients. As decreased stride and pace are associated with an increased risk of falls, further studies are warranted to evaluate the effectiveness of monitoring gait with the proposed reliable and valid system to reduce fall risk in hemodialysis patients.

## Figures and Tables

**Figure 1 jpm-13-01181-f001:**
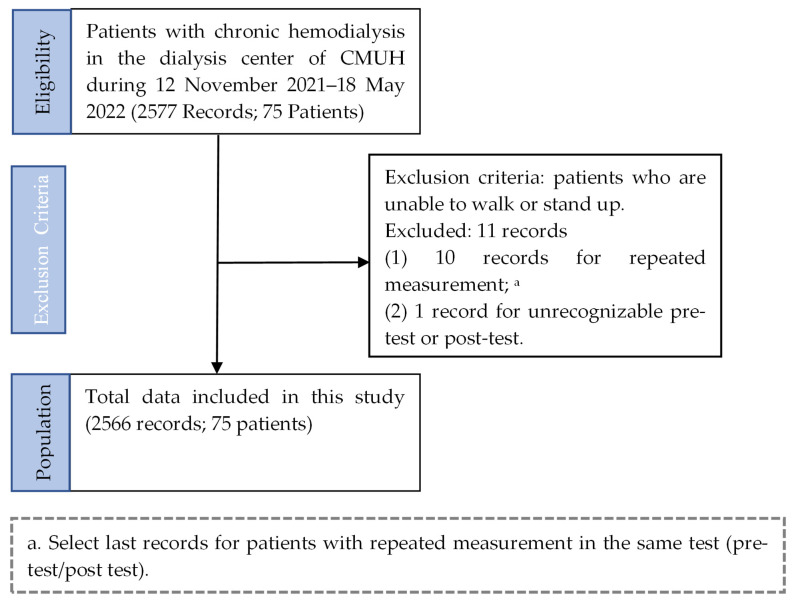
Flowchart of patient selection.

**Figure 2 jpm-13-01181-f002:**
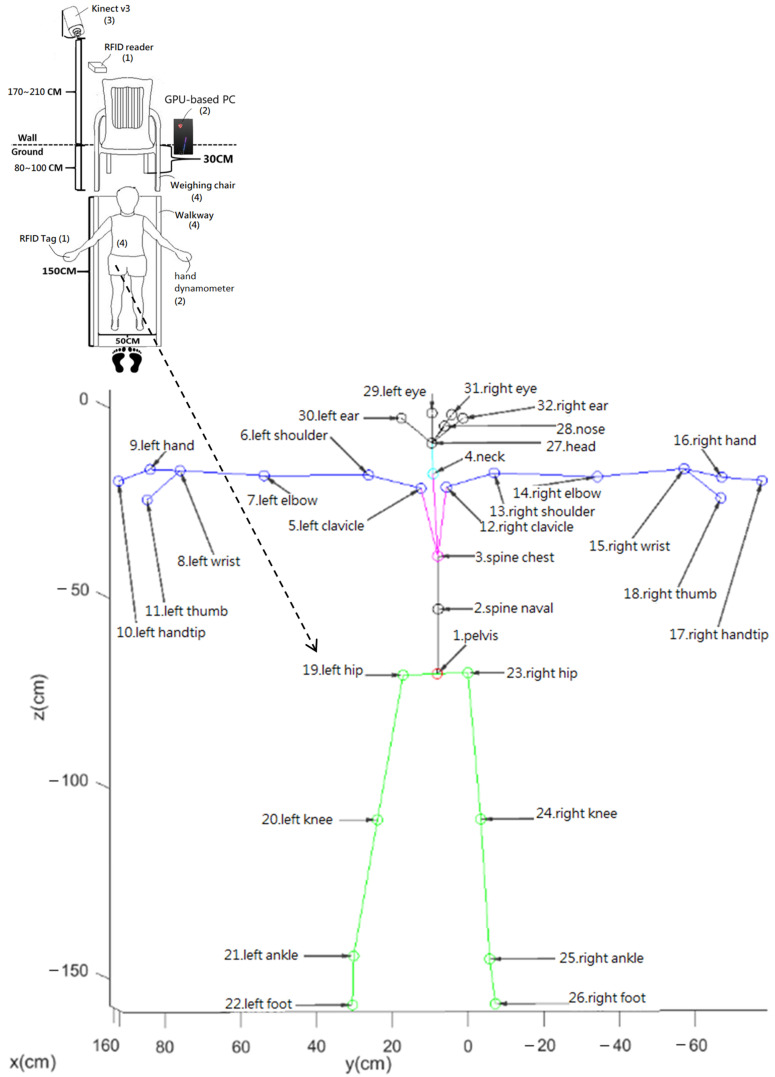
Four suggested workflow steps of K3S; Kinect v3′s 32 virtual skeleton joints are labeled No. 1–No. 32.

**Figure 4 jpm-13-01181-f004:**
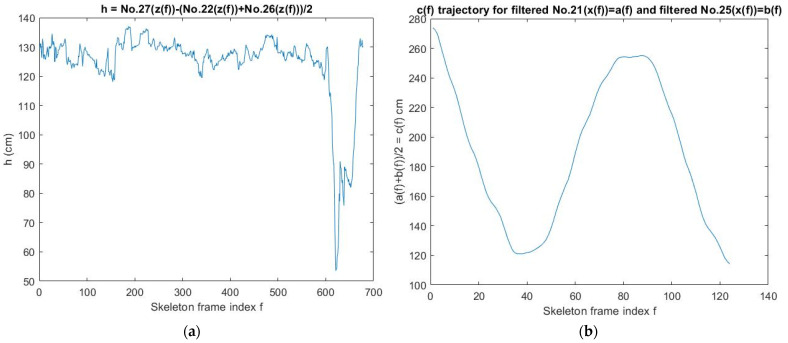
K3S experiments encountered some unreasonable movements and their solvers: (**a**) Suddenly stopping and bending over to pick up some objects. (**b**) Walking back and forth. (**c**) Sitting in a wheelchair to move on the walkway. (**d**) The result of one-way trajectory detection for the walking case **b**. (**e**) Standard stride length of the right foot. (**f**) A generalized stride length of feet is a modified version extended from **e** for patients.

**Figure 5 jpm-13-01181-f005:**
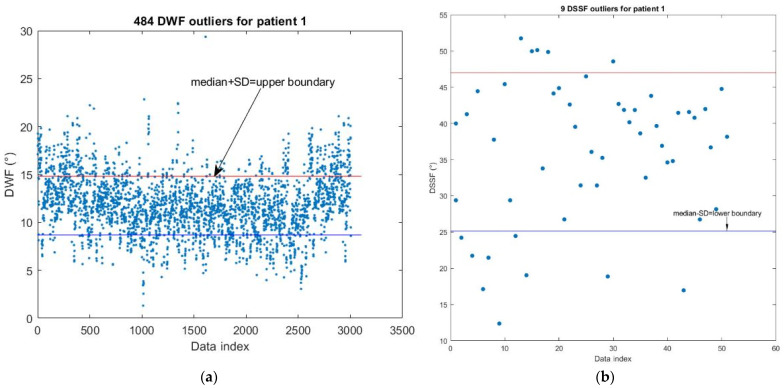
Detection of the personal DWF and DSSF outliers for patient 1 is based on (**a**) DWF boundaries for patient 1. (**b**) DSSF boundaries for patient 1.

**Figure 6 jpm-13-01181-f006:**
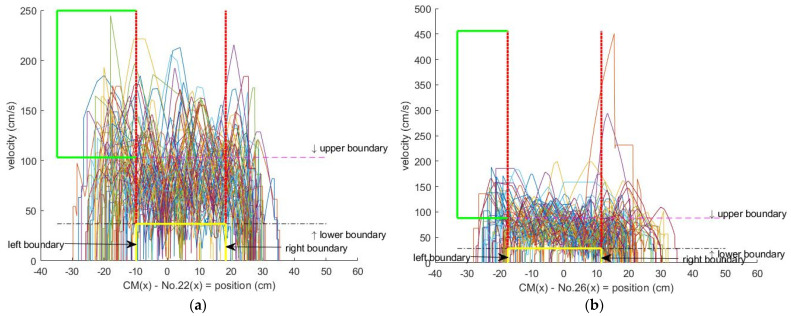
Detection of the personal unbalanced gait of patient 1. (**a**) Outliers in green and yellow boxes for the phase portrait of CM(x)−No.22(x). (**b**) Outliers in green and yellow boxes for the phase portrait of CM(x)−No.26(x).

**Figure 7 jpm-13-01181-f007:**
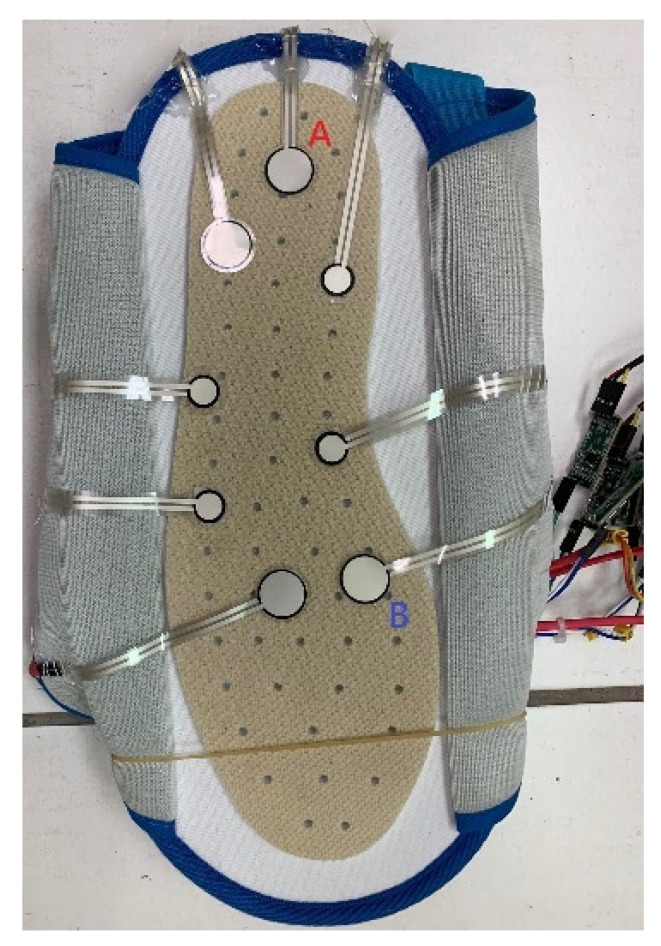
Two sensors (A and B) of a pressure insole.

**Figure 8 jpm-13-01181-f008:**
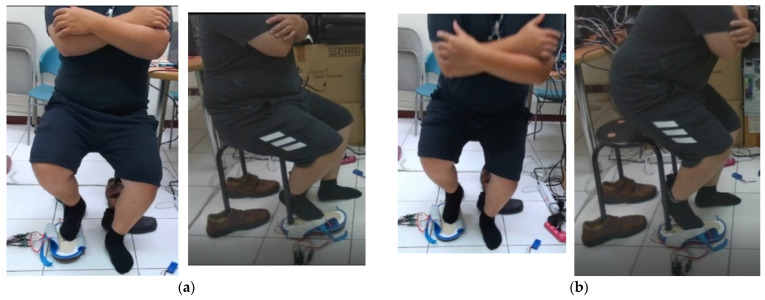
Calibration design of the sit-and-stand time (SST) algorithm: (**a**) Seated photo of male subject 2 (the current front and side view), (**b**) standing photo of the same participant (the current front and side view), and t(B) time value obtained after standing up, standing, and just leaving pressure sensor B. (**c**) The peak and valley positions of the pelvis z(t) curve measured by K3S calculated by the rise time algorithm for the Kinect v3 origin of the coordinate system. (**d**) The pelvis z position of the sitting posture and its time point “o” calculated by the standing up time algorithm, and the starting point “Δ” of the pelvis z position of the just completed standing and walking posture. (**e**) The pressure sensor A is installed under the foot of the chair; the pressure sensor B is the point stepped on when standing up; assume time t; B pressure curve is t(B) at the end of the pressure after being compressed; A pressure curve is at the beginning of the pressure. When it is falling, it is t(A), then the time for a single rise is SST≡ t(B)−t(A). (**f**) The average sampling time Δt is approximately 0.02 s.

**Figure 9 jpm-13-01181-f009:**
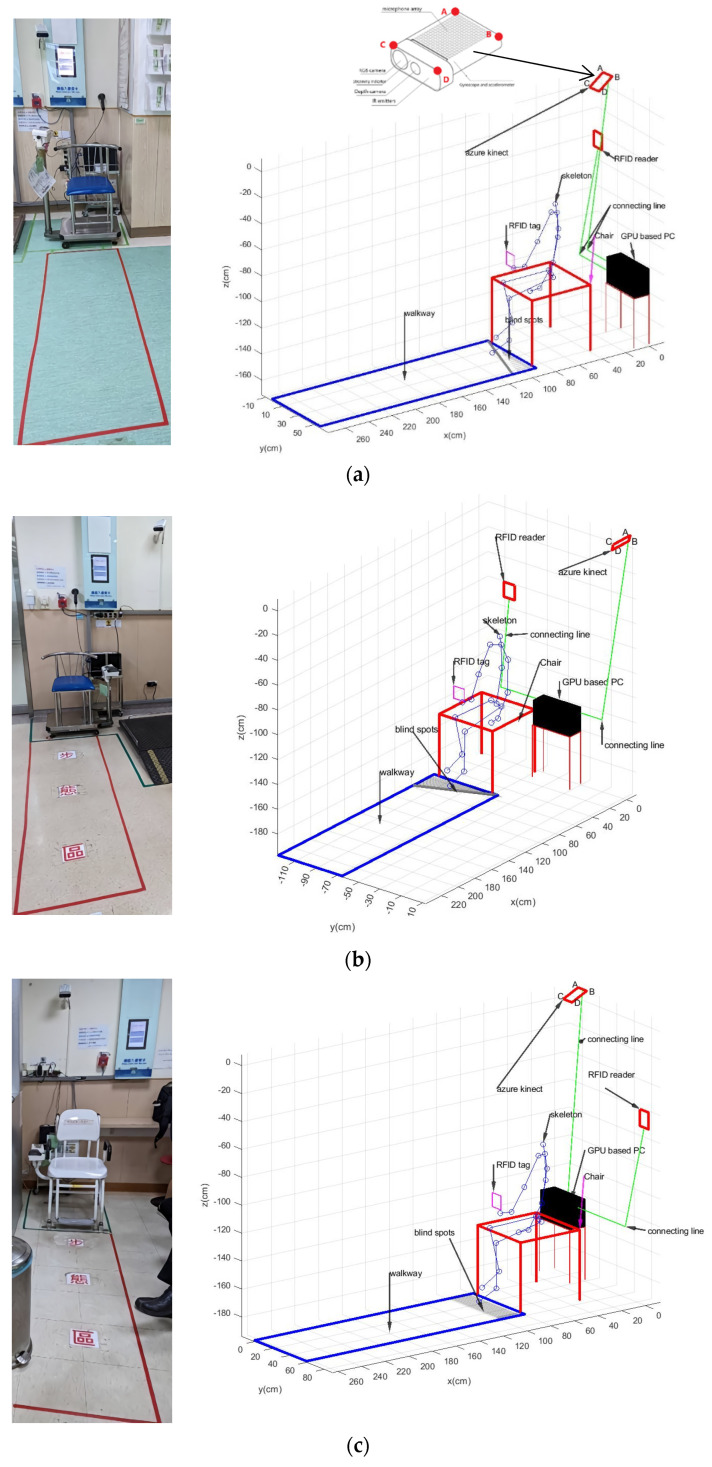
A clinical trial using three locations of the Dialysis Center of China Medical University Hospital (CMUH) was implemented as a unified coordinate system of the three walkways (**a**–**c**) was obtained after coordinate transformation.

**Figure 10 jpm-13-01181-f010:**
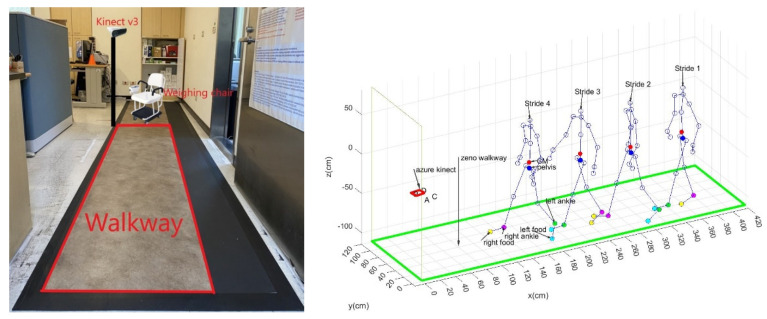
Zeno 426.72 cm × 121.92 cm walkway measures the stride and pace data analyzed by PKMAS software, and its spatial resolution is 1.27 cm. Calibration setting for the four strides and their paces measured by the Zeno Walkway and K3S.

**Figure 11 jpm-13-01181-f011:**
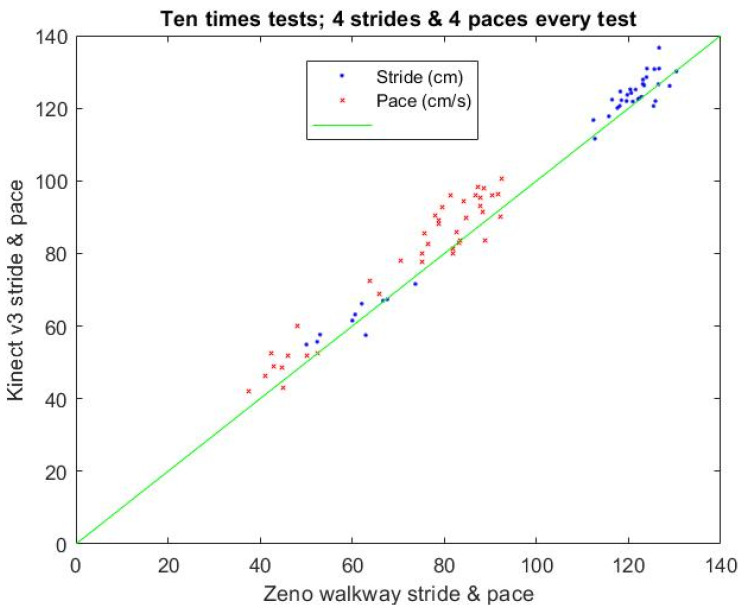
Comparison results for stride and pace measured using the Zeno Walkway and Kinect v3.

**Figure 12 jpm-13-01181-f012:**
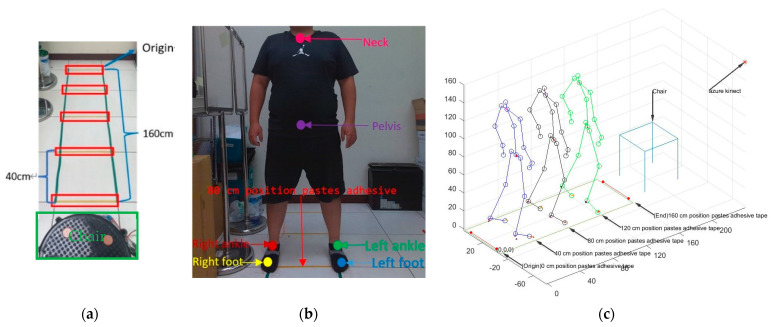
Stability analysis experiment design of the main virtual joints. (**a**) The adhesive tape lines on tiles of a short-distance 1.6 m walkway in the laboratory indicate that the size of a tile is a 40 cm square. (**b**) This participant was standing on an 80 cm position (which has an adhesive tape line). (**c**) This participant stands at three positions (40 cm, 80 cm, and 120 cm) which correspond to Kinect v3 virtual skeleton frames to check the stability of joints for the new origin of the coordinate system.

**Figure 13 jpm-13-01181-f013:**
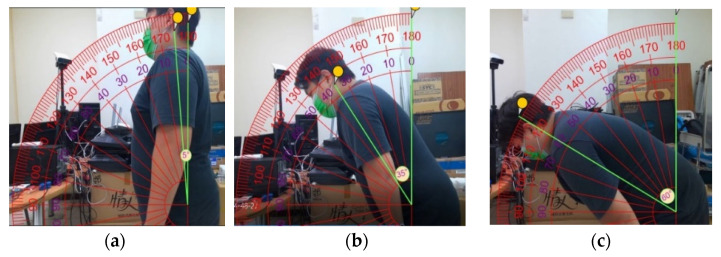
Reliability and validity analysis experiment of the actual forward angles (**a**) 5 degrees, (**b**) 35 degrees, and (**c**) 60 degrees of male subject 2.

**Figure 14 jpm-13-01181-f014:**
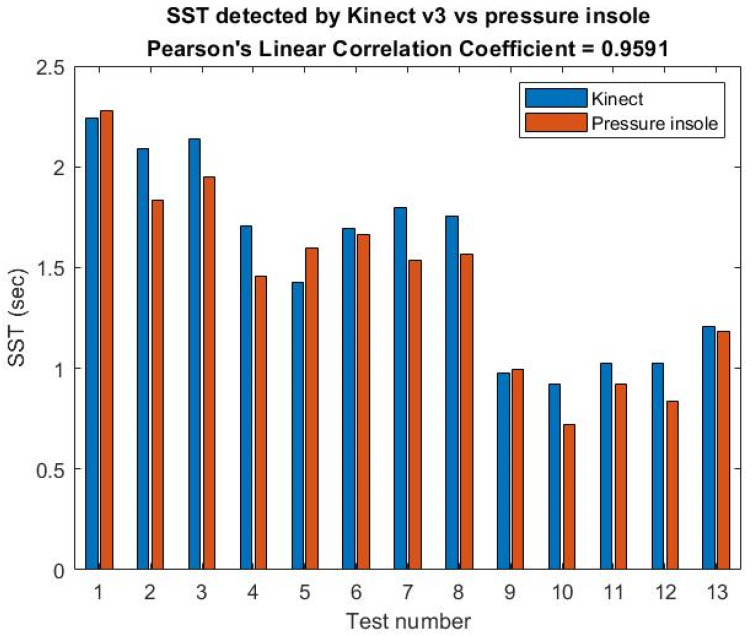
The SST calibration result is illustrated using the correlation of Kinect v3’s and two pressure sensors’ SST data.

**Figure 15 jpm-13-01181-f015:**
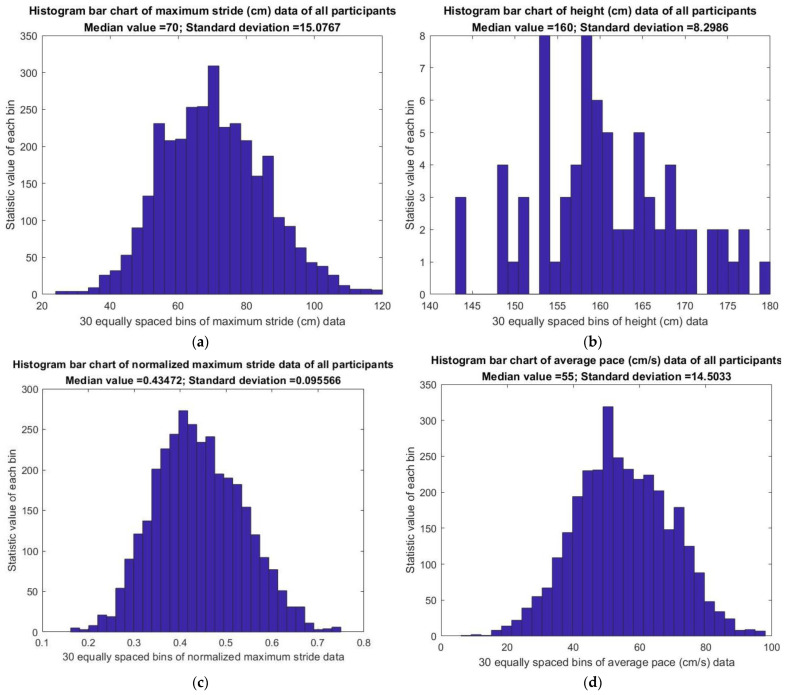
Analysis results of gait statistics of 75 patients (12 November 2021–18 May 2022) in this study: (**a**) Maximum stride length outliers are defined below the median–SD boundary. (**b**) The height histogram of all participants. (**c**) Normalized maximum stride length outliers are defined below the median–SD boundary. (**d**) Pace outliers are defined below the median–SD boundary. (**e**) SST outliers are defined beyond the median + SD boundary. (**f**) Shows the distributed numbers of the DWF outliers of all participants whose warning DWF outliers are defined beyond the median + SD boundary of these distributed numbers. (**g**) Shows the distributed numbers of the UPN of all participants whose warning UPN outliers are defined beyond the median + SD boundary of these distributed numbers.

**Table 1 jpm-13-01181-t001:** Baseline demographic data of 75 patients with chronic dialysis in the CMUH Dialysis Center. ^a^ Categorical variables are presented as numbers (frequency, %), and continuous variables are presented as medians (IQR). ^b^ Frequency was calculated by median per patient from 12 November 2021 to 18 May 2022.

Characteristics ^a^	Total Patients = 75
Age (years), median (IQR)	68.0 (59.5~72.3)
Age < 55 years, no. (%)	12 (16.0%)
55 ≤ Age <65 years, no. (%)	18 (24.0%)
65 ≤ Age <70 years, no. (%)	14 (18.7%)
Age ≥ 70 years, no. (%)	31 (41.3%)
Male, no. (%)	49 (65.3%)
Pre-test	
Total records, no.	1461
Frequency per patient ^b^, median (IQR)	13.0 (10.0~22.0)
Post-test	
Total records, no.	1115
Frequency per patient ^b^, median (IQR)	12.5 (8.0~20.5)

**Table 2 jpm-13-01181-t002:** Ai, Bi, Di, Gi, and Ci parameters used in the center of mass (CM) of K3S calculation.

**Ai**	**Joint No.**	**Bi**	**Joint No.**	**Di**	**Di Value**
A1	27	B1	4	D1	47.10%
A2	4	B2	3	D2	48.55%
A3	3	B3	1	D3	57.55%
A4	23	B4	24	D4	55.25%
A5	19	B5	20	D5	55.25%
A6	24	B6	25	D6	59.10%
A7	20	B7	21	D7	59.10%
A8	25	B8	26	D8	52.75%
A9	21	B9	22	D9	52.75%
A10	13	B10	14	D10	56.15%
A11	6	B11	7	D11	56.15%
A12	14	B12	15	D12	64.25%
A13	7	B13	8	D13	64.25%
A14	15	B14	16	D14	53.15%
A15	8	B15	9	D15	53.15%
**Link Name**	**Start Joint No.**	**End Joint No.**	**Gi**	**Ci**	**Ci Value**
Head and neck	27	4	g1	C1	8.41%
Upper trunk	4	3	g2	C2	16.67%
Lower trunk	3	1	g3	C3	27.35%
Thigh (right)	23	24	g4	C4	14.14%
Thigh (left)	19	20	g5	C5	14.14%
Shank (right)	24	25	g6	C6	4.05%
Shank (left)	20	21	g7	C7	4.05%
Foot (right)	25	26	g8	C8	1.36%
Foot (left)	21	22	g9	C9	1.36%
Upper arm (right)	13	14	g10	C10	2.54%
Upper arm (left)	6	7	g11	C11	2.54%
Forearm (right)	14	15	g12	C12	1.19%
Forearm (left)	7	8	g13	C13	1.19%
Hand (right)	15	16	g14	C14	1.06%
Hand (left)	8	9	g15	C15	1.06%

**Table 3 jpm-13-01181-t003:** Outliers in each gait parameter (12 November 2021–18 May 2022). ^a^ Frequency of outliers in each gait parameter. Abbreviations: P1, 1st percentage; P5, 5th percentage; P95, 95th percentage; P99, 99th percentage.

Parameters	Maximum Stride (cm) Total (N = 2566 Records)	Average Pace (cm/s) Total (N = 2566 Records)	SST (s) Total (N = 2566 Records)
Mean ± SD	87.9 ± 16.3	56.0 ± 17.4	4.1 ± 3.6
Fifth and 95th percentage			
Median (P5, P95), cm	86.7 (63.7, 116.1)	56.7 (27.6, 83.4)	3.2 (1.2, 10.2)
No. of records ≤P5 ^a^	130	130	129
No. of records ≥P95 ^a^	132	132	129
First and 99th percentage			
Median (P1, P99), cm	86.7 (48.1, 129.9)	56.7 (3.9, 94.2)	3.2 (0.8, 17.6)
No. of records ≤P1 ^a^	26	27	26
No. of records ≥P99 ^a^	26	26	26

**Table 4 jpm-13-01181-t004:** Number of records with outliers in any of the 3 gait parameters. ^a^ Total records are records with dialysis in the CMUH from 12 November 2021 to 18 May 2022. ^b^ Percentage = (Records of outlier/Total records).

Characteristics	Total Records ^a^	Parameters ≤P5 or ≥P95 Records (%) ^b^	Parameters ≤P1 or ≥P99 Records (%) ^b^
Total	2566	858 (33.3%)	192 (7.5%)
Dialysis month			
November 2021	21	10 (47.6%)	1 (4.8%)
December 2021	178	51 (28.7%)	16 (9.0%)
January 2022	262	89 (34.0%)	11 (4.2%)
February 2022	243	64 (26.3%)	12 (4.9%)
March 2022	299	102 (34.1%)	26 (8.7%)
April 2022	845	289 (34.2%)	65 (7.7%)
May 2022	718	253 (35.2%)	61 (8.5%)
Dialysis section			
Morning	1538	483 (31.4%)	108 (7.0%)
Afternoon	1028	375 (36.5%)	84 (8.2%)

**Table 5 jpm-13-01181-t005:** Gait parameters for 2566 records of pretest and posttest (comparison before and after dialysis). ^a^ (*p* value estimate): *p* values are calculated using the Wilcoxon rank sum test.

Characteristics	Pre-Test	Post-Test	*p* Value ^a^
	(Total Records = 1454)	(Total Records = 1112)	
Maximum stride (cm)			<0.001
Median (IRQ)	88.8 (79.7, 99.8)	83.5 (74.3, 96.1)	
Mean (SD)	90.2 (15.7)	85.0 (16.7)	
Min	29.3	24.5	
Max	146.3	138.2	
Average pace (cm/s)			0.409
Median (IRQ)	57.1 (45.0, 68.1)	55.8 (44.7, 67.5)	
Mean (SD)	56.2 (17.8)	55.7 (16.9)	
Min	0	0	
Max	116.8	102.7	
SST (s)			<0.001
Median (IRQ)	2.8 (1.9, 4.5)	3.6 (2.3, 5.7)	
Mean (SD)	3.8 (3.4)	4.6 (3.7)	
Min	0.1	0.1	
Max	42.4	59.0	

## Data Availability

The data underlying the results presented in the study are available from China Medical University Hospital. The data are owned by a third party and authors do not have permission to share the data.

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
