# Peer review of "Validation of Gait Measurements on Short-Distance Walkways Using Azure Kinect DK in Patients Receiving Chronic Hemodialysis"

_jpm, 2023, doi:10.3390/jpm13071181_

Round 1

Reviewer 1 Report

 This study aims to evaluate the effectiveness of a gait monitoring system adapted to the environmental conditions of medical practice.

The author reports on developing a system for chronic dialysis patients that reduces the physical space used by conventional gait monitoring systems and still allows for simple measurements, and evaluates the system's effectiveness. In addition, the author discusses the potential for future development of this system.

This study is very significant because it identifies the possibility of making effective use of data obtained from gait monitoring. The study's methodology, measurement system setup, statistical analysis, etc., are clearly stated and implemented.

Major Comments

The subjects of this study were 75 chronic dialysis patients. The reason for limiting the study subjects to chronic dialysis patients is unclear. In my opinion, selecting research subjects is the backbone of the study, and the reasons for the selection should be clearly stated. In addition, I believe the criteria for excluding research subjects should also be clearly stated. The exclusion criteria should also be indicated in the study flow chart (Fig. 1: L98-99). If any study subjects were excluded, it should be indicated.

In the Discussion part, the author describes how to correct short-distance walking (1.5 m) compared to long-distance walking (4 m). The author proposes normalization using the height of the study subjects. Although there are many individual factor parameters for normalization, it should be clarified why it was limited to height and why other individual factor parameters were not used.

Minor Comments

In Table 1 (L101-105), the letters and numbers overlap and are difficult to read. I think it would be simpler to correct them to facilitate understanding.

In Table 4 (L507-511), the letters and numbers overlap and are difficult to read. I think it would be simpler to correct them to facilitate understanding.

Author Response

We wish to express sincere gratitude to you for your constructive comments and helpful suggestions which lead to improvements of this paper. The responses to your comment are in the attached file.

Reviewer 2 Report

I have reviewed the manuscript by Zhi-Ren Tsai and coworkers, tilted “Validation of gait measurements on short-distance walkways using Azure Kinect DK in patients receiving chronic hemodialysis.  The following are my comments and recommendations to the authors:

Tsai and coworkers provide a very detailed description and analysis of a new approach to gait measurements that is applicable within a dialysis population.  The authors provide sufficient details of their procedure such that others could reproduce this approach.  It is likely but not yet proven that this approach will provide important clinical measures.

However, as in any study of a new diagnostic test (or a new application/ adaptation / approach of an older diagnostic test) certain information is expected.  This reviewer would recommend that the authors include some additional data from the current study that is not reported in the current manuscript.

1.      It would be important to know a bit more about the study population.  Given that the measures require that patients perform physical activity, it is important to be able to anticipate the barriers and limitations to the application of this test regimen.  Does the population include individuals who have limited mobility by virtue of age, BMI, comorbid conditions?  Table 1 includes some of the demographics.  The number of records is listed but this does not indicate if measures of gait were made for each of these records.  How many full gait measures or how frequently were the Kinect v3 gait measures made? If the gait measure were repeated, were the findings reproducible; was there a learning process that altered the scores?

2.     Did the cohort of patients include those with amputations with prostheses?  Were there any principally wheelchair or walker dependent patients?

3.     Are there data on patient acceptance or how well the measures were tolerated?  Any survey of the patient experience with the test measures/ acceptance of this additional burden?  Any need to have specially trained physical therapist or the like to help patients how to perform the tests properly? How many patients completed all of the tests? 

4.     How well did the new Kinect v3 measures agree with the prior accepted measures on individual patients?  This question speaks to the principle that if one is advocating for a new testing approach/ algorithm that one should be able to report how well it reiterates the gold standard method.  Do the authors have any comparator data?  This reviewer would argue that such data are not absolutely necessary if the new measure is reliable, reproducible and predictive of future outcomes.  The authors present a very detailed and careful analysis that addresses the reliability and reproducibility of their measure.

5.     This reviewer would like to believe that this measure will be predictive of important health milestones including fall risk, quality of life, and long-term survival by recognizing early impairment and allowing for treatment to improve these outcomes.  The current manuscript does not, however, provide any direct evidence of the clinical impact of the application of this measure and the changes in gait and mobility with any specific interventions.  This reviewer would argue that this will require additional evidence and is worthy of future investigation.  The current manuscript provides important evidence of a potentially significant advance in the longitudinal assessment of gait and mobility status and of known modulators of further impairment progression. This reviewer would recommend qualifying the conclusions to better acknowledge that the clinical benefits of the new approach remain to be fully demonstrated.

This reviewer wishes to acknowledge some specific strengths of this manuscript:

1.     Clear and detailed description of the measure, how to conduct the study on patients, and its reproducibility and reliability, including a detailed analysis of outliers etc. 

2.     Excellent illustrations and tables to provide the reader with the important data and their interpretation.

3.      A clear review of prior data and a solid evidence-based justification for the novel approach that these investigator report on in this manuscript.

Really only a few errors in syntax that need editing.  Very minor, mostly well written.

Author Response

(The authors gave the same response as above.)
